# Urinary Metabolome Analyses of Patients with Acute Kidney Injury Using Capillary Electrophoresis-Mass Spectrometry

**DOI:** 10.3390/metabo11100671

**Published:** 2021-09-30

**Authors:** Rintaro Saito, Akiyoshi Hirayama, Arisa Akiba, Yushi Kamei, Yuyu Kato, Satsuki Ikeda, Brian Kwan, Minya Pu, Loki Natarajan, Hibiki Shinjo, Shin’ichi Akiyama, Masaru Tomita, Tomoyoshi Soga, Shoichi Maruyama

**Affiliations:** 1Institute for Advanced Biosciences, Keio University, Tsuruoka 997-0052, Japan; hirayama@ttck.keio.ac.jp (A.H.); ra.benten.tsukasa.lab@gmail.com (A.A.); k-yushi@ttck.keio.ac.jp (Y.K.); yu-yu.k@ttck.keio.ac.jp (Y.K.); satsuki@ttck.keio.ac.jp (S.I.); mt@sfc.keio.ac.jp (M.T.); soga@sfc.keio.ac.jp (T.S.); 2Division of Biostatistics and Bioinformatics, Department of Family Medicine and Public Health, University of California San Diego, La Jolla, CA 92093, USA; bkwan@health.ucsd.edu (B.K.); mpu@health.ucsd.edu (M.P.); lnatarajan@health.ucsd.edu (L.N.); 3Department of Nephrology, Nagoya University Graduate School of Medicine, Nagoya 466-8560, Japan; shinjo14@nagoya2.jrc.or.jp (H.S.); akiyama.med@gmail.com (S.A.); marus@med.nagoya-u.ac.jp (S.M.)

**Keywords:** AKI, capillary electrophoresis-mass spectrometry (CE-MS), biomarker, urine

## Abstract

Acute kidney injury (AKI) is defined as a rapid decline in kidney function. The associated syndromes may lead to increased morbidity and mortality, but its early detection remains difficult. Using capillary electrophoresis time-of-flight mass spectrometry (CE-TOFMS), we analyzed the urinary metabolomic profile of patients admitted to the intensive care unit (ICU) after invasive surgery. Urine samples were collected at six time points: before surgery, at ICU admission and 6, 12, 24 and 48 h after. First, urine samples from 61 initial patients (non-AKI: 23, mild AKI: 24, severe AKI: 14) were measured, followed by the measurement of urine samples from 60 additional patients (non-AKI: 40, mild AKI: 20). Glycine and ethanolamine were decreased in patients with AKI compared with non-AKI patients at 6–24 h in the two groups. The linear statistical model constructed at each time point by machine learning achieved the best performance at 24 h (median AUC, area under the curve: 89%, cross-validated) for the 1st group. When cross-validated between the two groups, the AUC showed the best value of 70% at 12 h. These results identified metabolites and time points that show patterns specific to subjects who develop AKI, paving the way for the development of better biomarkers.

## 1. Introduction

Acute kidney injury (AKI) is defined as a severe decline in kidney function within a few hours. It is caused by various factors, including decreased blood flow during a major surgery or pain medication overuse. Because it is associated with morbidity and mortality, understanding the mechanisms underlying AKI progression and identifying body fluid biomarkers are urgently needed to develop medications for patients with AKI. To this end, several studies have investigated the molecular profiles of blood and urine during the progression of AKI. A change in serum creatinine is currently the gold standard marker for the onset of AKI as it is strongly associated with the filtering function of the kidney. In fact, the Kidney Disease: Improving Global Outcomes (KDIGO) criteria are based on serum creatinine levels and urine outputs [1]. However, changes in the serum level often occur after kidney injury. Therefore, significant efforts have been made to explore possible biomarker candidates that predict AKI earlier. For example, the levels of NGAL [2], cystatin C, TIMP-2, KIM-1 and IGFBP7 have been shown to correlate with the disease [3].

Recently, omics technologies have been applied to elucidate more comprehensive molecular profiles of the disease. The conducted studies differ not only by the body fluid type (blood or urine) but also by target molecules and measurement instruments used. The investigated molecular profiles and instruments include proteomes (LC-MS/MS, [4,5]), peptides (CE-MS, [6,7]) and metabolites (NMR, [8,9,10]; LC-MS, [11,12,13]; LC-MS and GC-MS, [14]; NMR and LC-MS/MS, [15,16]). Because AKI is a heterogeneous disease, these studies have been conducted in various populations and clinical settings, such as patients after cardiac surgery, those who received cisplatin Rx and critically ill children [17]. Thus, the molecular profiles of patients with AKI have been investigated from various points of view.

Similarly, we focused on patients admitted to an intensive care unit (ICU) after invasive surgery. Most patients recover in the ICU after major surgeries of organs, such as the brain and heart, but some patients develop AKI. The primary causes include ischemia during surgery, sepsis, drug-induced renal dysfunction and malignant tumors, and the disease is usually associated with the dysfunction of other organs and complications. In our present work, we conducted a comprehensive metabolome profiling of urine samples from patients with AKI. LC-MS has been one of the most widely used instruments for metabolome analyses. On the other hand, our institute has been developing CE-MS as a pioneer for twenty years [18]. It has been shown to more effectively quantify charged metabolites including amino acids [19], compared with other instruments, such as LC-MS and has been successfully applied to metabolome analyses of biological samples from patients with various diseases, including different liver diseases [20], colorectal cancer [21] and diabetic nephropathy [22]. We therefore employed CE-MS for analyzing urinary metabolites to take an advantage of our expertise.

Urine samples were collected at six time points from patients who underwent surgery. To identify metabolites potentially associated with the progression of the disease, the metabolite levels in patients who developed AKI were compared with those who did not. Our study may provide targets for further biological studies to investigate disease progression mechanisms or potential biomarkers for the early detection of AKI.

## 2. Results

### 2.1. Comprehensive Differential Analysis of Metabolites among AKI and Non-AKI Subjects

Urine samples were collected from patients in Nagoya University Hospital who were admitted to the ICU after invasive surgery. Urine samples were collected at six time points, including before surgery (Pre, time point A), on admission to the ICU (0 h, time point B) and 6 (time point C), 12 (time point D), 24 (time point E) and 48 h (time point F) after admission to the ICU (Figure 1a).

We first selected 61 patients, designated as the 1st group, for the measurement of metabolome profiles. After admission, 24 and 14 developed mild and severe AKI, respectively (designated as mild AKI and severe AKI). The remaining 23 patients did not develop AKI (non-AKI). The 2nd group consisted of 20 mild AKI and 40 non-AKI patients (see the Materials and Methods for further details regarding patient selection). The changes in the metabolomic profiles over the six time points were analyzed using CE-TOFMS. We quantified 512 known metabolites and 155 uncharacterized peaks. The metabolite profiles of the two groups were measured on different days in different batches.

To investigate differences in the level of metabolites between AKI (both mild and severe) and non-AKI patients at any time point, we calculated the log2-ratio of metabolite levels in AKI compared with non-AKI patients in the 1st group (Figure 1b left for known metabolites, Appendix A left for all metabolites). We observed a cluster of metabolites with low levels in AKI compared with non-AKI patients at 6, 12 and 24 h in the 1st group (Figure 1b left, corresponding cluster of metabolites colored in red). For the same set of metabolites, we observed a similar decrease in their levels at 6 and 12 h in the 2nd group, although the trend was less clear. Principal component analysis (PCA) revealed that the metabolome profiles of the three subject categories, including non-AKI, mild AKI and severe AKI, were separated at 24 h in the 1st group (Figure 2). No clear separation was observed for the rest of the time points (Appendix A).

We analyzed the differences in metabolite levels between non-AKI, mild and severe subjects in the 1st group for all metabolites and all time points except 48 h. The 48-h time point was excluded from this analysis because one of our purposes was to identify early biomarkers. In addition, we thought that metabolites from 48 h may have significant downstream effects on the disease onset. We conducted multiple testing corrections for all metabolites and time points. The results are shown in Table 1. After multiple testing corrections, no statistically significant differences were observed among the three subject categories at Pre (A), 0 h (B) or 6 h (C). However, at 12 h (D), glycine (Gly) was significantly reduced in subjects with mild or severe AKI compared with those without AKI (ANOVA, BH corrected *p* < 0.05). At 24 h (E), the levels of five metabolites were significantly different between the three subject categories (ANOVA, BH corrected *p* < 0.05). For these metabolites, no correlation was observed between their level and the severity of disease, except for uncharacterized metabolite AU034, which showed a difference between mild AKI and severe AKI. Of note, we did not find hard-to-interpret cases in which the median metabolite level in non-AKI patients was between those of the other two categories of AKI (mild and severe). The top three metabolites, including ethanolamine, glutamine (Gln) and glycine (Gly), are marked in the heatmap of Figure 1b, all of which belong to the cluster colored in red.

The raw metabolite levels (µM) were normalized by urinary creatinine levels (g/L). Thus, urinary creatinine levels impacted the statistical analysis results. We analyzed the non-normalized metabolite levels for the top three metabolites shown in Table 1 to determine if the statistically significant differences in these metabolites remained (Appendix A).

In the analyses above, each time point was treated independently. To take time-series patterns of metabolite levels for each subject into account, we modeled the patterns using two linear mixed effects models. Using these models, we then tested whether there was any difference in the patterns among the three categories of subjects in the 1st group (for details regarding statistical models, see the Materials and Methods).

In the first model, the metabolite level is assumed to be influenced by disease states, time points and subjects. Using this model, we tested whether disease states actually affected the metabolite level (Table 2). We found eighteen metabolites with BH corrected *p* values < 0.05. The top three metabolites in Table 1 were also included in Table 2.

In the second model, the metabolite level is assumed to be influenced by disease states, time points and subjects. In addition, changes in the level of metabolites over the time points are assumed to be influenced by the disease state. We tested whether disease states actually affected the slopes of the changes in metabolite levels over the time points (Table 3). We found eight metabolites with BH corrected *p* values < 0.05, piperidine being the most significant. These significant metabolites appeared to be substantially different from those shown in Table 2, indicating orthogonality of metabolite levels and their degree of change over the time points.

The patterns in the changes in ethanolamine, glutamine (Gln) and glycine (Gly) over the time points in the 1st group of patients are shown in Figure 3a. We observed slight differences in metabolite levels among non-AKI and AKI patients at 6 h, which became statistically significant at 12 h or 24 h. This significance was not reproduced in the 2nd group (Figure 3b, middle panel; for example, glycine). However, after removing subjects who were diagnosed with AKI based on the serum creatinine level within two days of ICU admission, we found a statistically significant difference for glycine.

### 2.2. Measurement of uNGAL

Urinary NGAL (uNGAL) is a previously studied protein marker for AKI. We investigated uNGAL levels in the 1st group of subjects at 12 h and 24 h, the two time points that showed significant differences in metabolites between non-AKI and AKI patients. We observed substantial upregulation of NGAL in three subjects (extreme outliers) with AKI at both time points (Figure 4a). In addition, at 24 h, the subjects with AKI showed slight upregulation of NGAL compared with the non-AKI group. However, the difference in NGAL levels between AKI and non-AKI was not statistically significant. We also tested the performance of AKI prediction using NGAL levels at each time point. The AUCs (areas under the curves) at 12 h and 24 h were below 0.7 (Figure 4b), indicating an unremarkable prediction performance. We performed the same analyses using uNGAL without normalization by creatinine and found similar results (Appendix A).

### 2.3. Prediction of AKI Based on Metabolome Profiles and Clinical Variables Using LASSO

Finally, to determine which metabolites or clinical variables effectively predict AKI, we used LASSO, which automatically selects a set of metabolites and clinical variables for the prediction of disease states. It further generates a linear equation for predicting the states based on the selected metabolites and clinical variables. We used metabolome profiles at specific time points and four clinical variables, including age, gender, surgery duration and baseline serum creatinine level, as inputs and non-AKI or AKI (mild or severe) as outputs (targets).

Based on the AUCs in the test sets (¼ of the subjects in the 1st group), predictions based on metabolome profiles at Pre, ICU, and 6 h in the 1st group of subjects (training set, ¾ of the subjects) did not show high performance (Figure 5a, left boxplots of plot pairs, designated as within 1st group cross-validation). However, at 12 h, the median AUC was 74%, showing moderate performance (Figure 5b, left). At 24 h, the median AUC increased to 89% (Figure 5c, left). The top ten automatically selected variables based on the variable selection rate (designated as V.S.R.) are shown in the right panel of Figure 5b,c. Glycine and ethanolamine were the most frequently selected variables at 12 h and 24 h, respectively. Because the concentrations of these two metabolites were relatively high (median metabolite levels were 390 and 115 µM/creatinine g/L for glycine at 12 h and ethanolamine at 24 h, respectively), they had a major impact on the prediction. Their coefficients were both negative, indicating that subjects with low levels of these metabolites will likely develop AKI. Consistently, both metabolites showed downregulation in patients with AKI (Figure 3). Most of the other metabolites appeared to have a minor effect that might slightly improve the performance of AKI prediction.

We also investigated the performance of predictions when the 1st and 2nd groups were used as training and test sets, respectively, for calculating AUCs (Figure 5a, right boxplots of plot pairs, designated as 1st to 2nd group cross-validation). The AUC at Pre and ICU admission were close to 50%, indicating a low prediction ability of the metabolome profiles at these two time points. However, the median AUC was 64% at 6 h and increased to 70% at 12 h. At 24 h, the AUC decreased to 63%, although the AUC of within 1st group cross-validation was as high as 89%, implying that 24 h showed a strong batch-specific effect.

We expected two clinical variables (baseline serum creatinine levels and duration of surgery) to be efficient predictors of AKI, but these variables were rarely selected by LASSO. In fact, there was no statistically significant difference in baseline serum creatinine levels or surgery durations among non-AKI, mild and severe AKI patients (Appendix A).

## 3. Discussion

In the present study, we analyzed the difference in metabolome profiles between subjects who developed AKI and those who did not. We observed differences in metabolic profiles between these two groups (Figure 1). Some metabolites showed statistically significant differences (Table 1, Table 2 and Table 3). For example, we observed the downregulation of ethanolamine around 24 h for patients with AKI (Figure 3a). There are previous studies that may be relevant to this observation. For instance, phosphoethanolamine inhibits mitochondrial respiration and reduces reactive oxygen species [5], which may protect the kidney. Because phosphoethanolamine is generated from ethanolamine by ethanolamine kinase, we speculate that changes in the ethanolamine level in our study affected the regulation of phosphoethanolamine, which contributes to renal protection. In our data, phosphoethanolamine was decreased in patients with mild or severe AKI at 12 h (*p* = 0.0395, ANOVA) and 24 h (*p* = 0.0457, ANOVA), although this result was not statistically significant after multiple testing corrections. Of note, Martin-Lorenzo et al. [15] showed that patients with AKI had increased urine levels of phosphoethanolamine compared with healthy donors within 48 h of diagnosis, at 7 days of follow-up and at nephrology discharge. However, the clinical setting in their study was substantially different from ours. Alternatively, plasma phosphoethanolamine has been shown to be decreased in patients with major depressive disorder [23]. Thus, the levels of ethanolamine, which is generated from phosphoethanolamine via phosphoethanolamine phosphatase, might be associated with the stress levels of patients after surgery.

The downregulation of glutamine and glycine around 12–24 h was also observed (Figure 3a). Beck et al. [24] reported that glycine is decreased in rat kidneys with ischemia, which is consistent with our results. Previous reports have described the protective effects of these two metabolites on the kidney. For example, Hu et al. [25] found that glutamine administered after the initiation of sepsis ameliorated AKI induced by sepsis in a mouse model. Similarly, Yim et al. [26] demonstrated that glycine has a protective effect on kidneys with ischemia in a rat model.

After multiple testing corrections, statistical analyses based on the mixed effects models provided more significant metabolites than the standard ANOVA, showing the advantage of including time-series metabolite patterns in the mixed effects model. However, our models assumed a linear rate of change in metabolite levels, and due to the small number of samples, we could not allow for between-subject variability in the slope of changes in metabolite levels.

Urinary NGAL (uNGAL) is a powerful candidate biomarker for AKI. uNGAL levels at 24 h were slightly higher in patients with AKI, although it was not statistically significant (Figure 4a right, *p* = 0.164, ANOVA). The performance of predicting AKI based on uNGAL was rather limited in our study population. It is difficult to compare this result with previous results because the study designs, including clinical settings and definitions of cases and controls, were not identical to ours. The AUC values for AKI prediction in general patients in the ICU based on their uNGAL appear to range between 55 and 80% [27]. Using patients with more focused clinical characteristics, the AUC may be considerably higher. In fact, children with AKI, who are less likely to have comorbidities, appear to show a more evident increase in NGAL compared with adults with AKI [28,29].

The significant difference in several metabolites between the AKI and non-AKI groups depended on whether the metabolite levels were from 1st or 2nd group, implying that metabolite levels were sensitive to the batch effect. Nevertheless, we observed a trend in the metabolite levels. Specifically, the metabolites downregulated in patients with AKI from the 1st group were also likely to be downregulated in those from the 2nd group (Figure 1b). Glycine showed a statistically significant difference between AKI and non-AKI in the 1st group but not in the 2nd group (Figure 3b). However, after removing patients with an early increase in serum creatinine, we found a statistically significant difference. Thus, the time at which serum creatinine level reaches the disease level affects the metabolome profile. Because these subjects have heterogeneous clinical backgrounds, selecting subjects with specific clinical characteristics may enhance the differences in metabolome profiles between AKI and non-AKI patients, and the statistical significance may be robust against batch effects. In addition, increasing the number of subjects and testing our hypothesis in an independent institute using independently prepared subjects are warranted.

The batch effect also affected the prediction performance of AKI with the 1st group as a training set and the 2nd group as a test set (Figure 5a). The batch effect was most evident at 24 h. In particular, separation of the metabolome profiles of three subject categories was apparent at 24 h (Figure 2). The cross-validated AUC within the 1st group was 89% at this time point but decreased to 63% for the 1st to 2nd group cross-validation. In contrast, a minimal batch effect on AKI prediction was observed at 12 h. The AUC values were approximately 70% for both within-1st-group cross-validation and 1st to 2nd group cross-validation. The performance was slightly better than that using uNGAL, and we suggest that prediction at 12 h provides the most conservative estimate of the prediction performance. Because the 2nd group consisted of non-AKI and mild AKI patients and did not include subjects with severe AKI, our prediction model showed some ability to discriminate patients who will develop mild AKI from those who will not develop AKI. Of note, the AUC values from the 1st to 2nd group cross-validation started to increase at 6 h, implying that metabolic profiles at 6 h do contain some information regarding the risk of AKI, and this was determined by our time-series metabolome profile analyses. By adding more subjects to the study and stratifying them based on their clinical variables, prediction performance may be improved [30].

The comparison of the AKI prediction performance in this study with that of previous research is not straightforward because study designs, clinical backgrounds of the participating subjects, protocols of the sample measurements and variables used for the predictions vary among studies. The AUCs of AKI predictions using candidate biomarkers in previous studies range from 50% to 100% [31,32]. Zacharias et al. [10] predicted AKI following cardiac surgery based on urinary metabolome profiles (NMR) of patients and identified that the AUC at 24 h after the surgery was 83%, which is comparable to the AUC at 24 h after ICU admission in our study (89%) using the 1st group of patients. Various machine learning techniques have been applied for AKI prediction [33], although the AUCs were usually below 80% when using biomarker candidates as inputs [31,34]. AKI prediction based on clinical variables may be less laborious than that based on body fluid biomarkers, but in our study, clinical parameters were not selected as efficient predictors by LASSO. With a considerably larger size of longitudinal health records, AKI was predicted with higher accuracy using an advanced machine learning approach [35].

Most metabolome analyses for patients with AKI have been conducted using LC-MS, GC-MS or NMR. In contrast, our study used CE-MS. This method has the advantage of detecting charged metabolites, which constitute a major proportion of the metabolome in organisms. Therefore, our analyses may have provided more accurate metabolome profiles of patients with AKI compared to previous studies that use other types of instruments, and thus, we believe that our time-series metabolome data provide valuable insight into the mechanisms by which the metabolome is regulated during the progression of AKI. Notably, from our CE-MS data, we observed uncharacterized peaks (such as AU034 and AU014 in Table 1) with differences among non-AKI, mild and severe AKI. Some uncharacterized peaks were selected by LASSO for the prediction of AKI. Previous work also discovered uncharacterized peaks from CE-MS that might reflect the pathophysiology of the kidney in diabetic patients [22]. Therefore, we suggest that these uncharacterized peaks may have potential to predict AKI after the assessment of their reliability [36].

## 4. Materials and Methods

### 4.1. Collection of Medical Information and Specimens from Patients

Urine samples were collected from patients in Nagoya University Hospital who were admitted to the ICU after invasive surgery in various departments (Table 4 and Table 5). We received informed consent from all participants, and urine samples were collected at six time points, as shown in Figure 1a.

These patients were classified into three groups: those who were not diagnosed with AKI (non-AKI), those who were diagnosed with mild AKI (mild AKI) and those who were diagnosed with severe AKI (severe AKI). Our criteria for discriminating the three subject categories were based on our previous study [37] and KDIGO [1]. In particular, the baseline serum creatinine (Cr) in each patient was measured for 30 days before admission to the ICU, and the highest Cr (peak Cr) among days one to seven after surgery was calculated. If the peak Cr was 1.50 to 1.99 times greater than the baseline, the patient was diagnosed with mild AKI. If it was ≥2.00 times the baseline, then the patient was diagnosed with severe AKI. For the remaining patients, if the maximum change in the creatinine level within 48 h compared with the baseline was ≥0.3 mg/dL, the patient was classified with mild AKI. Otherwise, the patients were non-AKI. We did not consider urine output for the discrimination.

First, urine samples from 61 initial patients (non-AKI: 23, mild AKI: 24, severe AKI: 14, designated as the 1st group) were measured, followed by measurement of urine samples from 60 additional patients (non-AKI: 40, mild AKI: 20, designated as the 2nd group). The subjects for metabolome analyses were selected considering the balances in ages and genders (Appendix A).

Clinical characteristics of the patients in the 1st and 2nd groups are shown in Table 4 and Table 5, respectively. Urine samples were stored at −80 °C until analysis.

### 4.2. Urine Creatinine (Cr) Measurements by an Enzyme Method

The concentrations of urinary metabolites may show drastic changes as they substantially increase upon sweating and are reduced by water intake. Therefore, metabolite concentrations are usually normalized to the urinary creatinine (Cr) level. Cr is the final product of creatine and is continuously synthesized non-enzymatically by the dehydration of creatine in muscle cells. Cr is a nitrogen compound in the blood and is filtered through the glomerulus. Most Cr is excreted in urine without being reabsorbed in tubules.

The urinary Cr level was also used to determine the dilution level of urine prior to metabolome analysis by CE-MS to adjust the metabolite concentrations, ensuring that they fall within the dynamic range of the measurement. The dilution level depending on the urinary Cr level is shown in Appendix A. After measurements, the original concentrations of the metabolites were calculated based on the dilution level.

We applied an enzymatic method to measure the Cr value following the procedure prepared by Toyobo Co. Ltd. [Enzyme reagent A: KTCN-802, Enzyme reagent A: KTCN-812, standard (concentration 5 mM): KTCN-001].

### 4.3. Preprocessing of Urine Samples

Urine samples stored at −80 °C were thawed at room temperature, and 20 µL were transferred to 1.5-mL tubes. Next, 2 mM of internal standards (IS) within Milli-Q water were added. For the cation mode, methionine sulfone and 3-aminopyrrolidine were used as ISs. For the anion mode, trimesate, D-camphor-10-sulfonic acid sodium salt (CSA) and 2-(N-morpholino)ethane sulfonic acid (MES) were used. Milli-Q water was added for dilution based on the urinary Cr to achieve a Cr concentration of approximately 10 mg/dL (Appendix A). The samples were transferred to tubes for ultrafiltration and centrifuged at 10,000× *g* and 4 °C for 30 to 50 min. Finally, 7 µL of each sample were transferred to vials for metabolome analyses.

### 4.4. Chemicals

Methionine sulfone was purchased from Alfa Aesar (Ward Hill, MA, USA), and hexakis-(2,2-difluoroethoxy)-phosphazene (Hexakis) was obtained from SynQuest Laboratories (Alachua, FL, USA). All other reagents were from Sigma-Aldrich (St. Louis, MO, USA) or Wako Pure Chemicals (Osaka, Japan). All chemicals were of analytical or reagent grade. Water was purified with a Milli-Q purification system (Millipore, Burlington, MA, USA).

### 4.5. Instruments and Analytical Conditions

CE-TOF-MS analysis was performed using an Agilent 7100 CE system (Agilent Technologies, Waldbronn, Germany), Agilent 6224 LC/MS TOF system, Agilent 1260 series isocratic HPLC pump, G1603A Agilent CE-MS adapter kit and G1607A Agilent CE-ESI-MS sprayer kit (Agilent Technologies, Santa Clara, CA, USA). For the analysis of anionic metabolites, the ESI sprayer was replaced with a platinum needle instead of the initial stainless steel needle [38]. The other conditions related to the CE-ESI-MS sprayer were unchanged. For system control and data acquisition, Agilent Chemstation software was used for CE, and Agilent MassHunter software was applied for TOF-MS. The analytical conditions of CE-TOF-MS have been described elsewhere [20]. Briefly, cationic metabolites were separated on a fused-silica capillary column (50-m internal diameter × 100-cm total length) filled with 1 M formic acid as the electrolyte. A new capillary was flushed with the electrolyte for 20 min, and the capillary was equilibrated for 4 min by flushing with the electrolyte before each run. The sample solution was injected at 5 kPa for 3 s (approximately 3 nL), and a positive voltage of 30 kV was applied. The sheath liquid, which was methanol/water (50% *v*/*v*) containing 0.1 M Hexakis, was delivered at 10 L/min. Anionic metabolites were separated using a commercially available COSMO(+) capillary column coated with a cationic polymer (Nacalai Tesque, Kyoto, Japan). Ammonium acetate solution (50 mM, pH 8.5) was used as the electrolyte [38]. Before the first use, a new capillary was flushed successively with the electrolyte, 50 mM acetic acid (pH 3.4), and then the electrolyte again for 20 min each. Before each run, the capillary was equilibrated by flushing with 50 mM acetic acid (pH 3.4) for 2 min and then with the electrolyte for 5 min. The sample solution was injected at 5 kPa for 30 s (approximately 30 nL), and a voltage of −30 kV was applied. The sheath liquid, which was ammonium acetate (5 mM) in methanol/water (50% *v*/*v*) containing 0.01 M Hexakis, was delivered at a rate of 10 L/min. The results were automatically recalibrated relative to the masses of the two reference standards in each mode. Cationic analysis used the 13C isotopic ion of the protonated methanol dimer (2MeOH + H)+, *m*/*z* 66.06306 and protonated Hexakis (M + H)+, *m*/*z* 622.02896, whereas the anionic analysis used the 13C isotopic ion of deprotonated acetic acid dimer (2CH_3_COOH − H)−, *m*/*z* 120.03834 and Hexakis + deprotonated acetic acid (M + CH_3_COOH − H)−, *m*/*z* 680.03554. Mass spectra were acquired at a rate of 1.5 cycles/s from *m*/*z* 50 to 1000.

### 4.6. Data Processing of CE-MS

We used proprietary software developed in our institute (MasterHands Version 2.17.3.18) for the peak detection and quantification of metabolites [39,40]. Urinary metabolite levels (UA) were calculated after normalization by creatinine (Cr) using the following equation:Urinary metabolite levels (UA) (µmol/mmol Cr)= (metabolite level × dilution factor)/(Cr × 10/113.118)
where 113.118 g/mol denotes the molecular mass of Cr.

### 4.7. Urinary NGAL Measurement

Urinary NGAL (uGAL) measurements were performed by SRL, Inc. (Tokyo, Japan).

### 4.8. Statistical Analyses

Most statistical analyses were conducted using the statistical programming language R. ANOVA was applied to calculate statistical differences in metabolite levels among non-AKI, mild AKI and severe AKI. Metabolite concentrations were log2-transformed. For a metabolite with a level below the detection limit, 0.5 × (minimum metabolite level among detected metabolite levels in the subjects) was assigned as their metabolite level for each subject. If the levels of a specific metabolite were below the detection limit for all the samples, then the metabolite was removed from further analyses. For multiple hypothesis test corrections, we used Benjamini–Hochberg (BH) and Bonferroni (BF) methods.

For modeling the time-series patterns of metabolites, we used the following two linear mixed effects models (Equations (1) and (2)) using the lme4 package of R.

Model 1: The level of a specific metabolite *m_ij_* in each subject *i* at occasion *j* was modeled as
*m_ij_* = *β*_0_ + *γ_i_* + *β*_1_ · *d_i_* + *β*_2_ · *t_j_* + *ϵ_ij_*(1)
where *β*_0_, *β*_1_ and *β*_2_ are the fixed effects for the intercept, disease state and time point, respectively, *γ_i_* is the random intercept for subject *i*, *d_i_* is the disease state for subject *i*, *t_j_* is the time (0, 6, 12, 24 or 48 h) at occasion *j*, and *ϵ_ij_* is the error for subject *i* at occasion *j*. *γ_i_* and *ϵ_ij_* are assumed to independently follow a normal distribution with unknown variances *σ*_1_^2^ and *σ*_2_^2^, respectively [i.e., *γ_i_* ~ *N*(0, *σ*_1_^2^), *ϵ_ij_* ~ *N*(0, *σ*_2_^2^)]. The corresponding null model was *β*_1_ = 0.

Model 2:*m_ij_* = *β*_0_ + *γ_i_* + *β*_1_ · *d_i_* + *β*_2_ · *t_j_* + *β*_3_ · *d_i_* · *t_j_* + *ϵ_ij_*(2)
where *β*_3_ is the coefficient for the interaction term of disease state and time. The corresponding null model was *β*_3_ = 0.

Analyses using the least absolute shrinkage and selection operator (LASSO) were conducted with the glmnet package of R to construct linear models to predict the disease status of patients with metabolome profiles and clinical variables. Cross-validation of the models was performed within the 1st group by randomly dividing the patients into training (¾) and test (¼) sets. The optimal parameter (λ) was selected through five-fold cross-validation within the training set using deviance as the loss function. This procedure was repeated 100 times. Alternatively, cross-validation was conducted using the entire 1st group as the training set and the entire 2nd group as the test set.

## 5. Conclusions

Our CE-MS-based time-series urinary metabolomic profile analyses identified metabolites and time points that show patterns specific to patients who develop AKI, and our statistical models and machine learning technique were efficient for the identification from large size of the metabolome data. Using an increased number of patients with similar clinical backgrounds, more accurate predictions may be possible, leading to the development of more efficient biomarkers for AKI.

## Figures and Tables

**Figure 1 metabolites-11-00671-f001:**
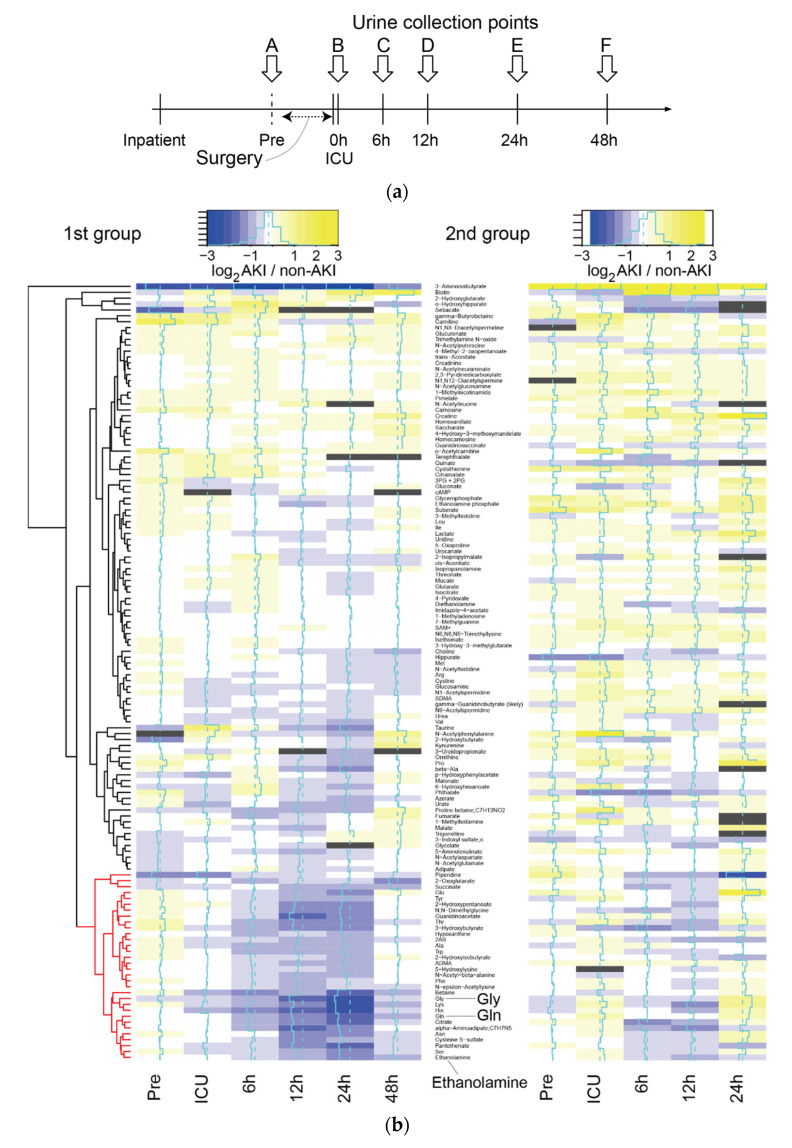
Overall trend in metabolome profiles. (**a**) Timeline of urine collection from inpatients. (**b**) Log2-ratio of the metabolome profile of AKI vs. non-AKI, represented by the color gradient from blue to yellow in the heat map. The metabolome profiles of the 2nd group (**right**) were listed in order based on the results of metabolome profile clustering in the 1st group (**left**). Dark gray represents an insufficient number of subjects. The cluster of metabolites showing downregulation in patients with AKI are marked in red. We did not have a sufficient number of samples at 48 h in the 2nd group, and they were omitted from the heatmap. The heatmap was generated using the heatmap.2 function in the gplots package of R, and layout and appearances were edited with Adobe Illustrator.

**Figure 2 metabolites-11-00671-f002:**
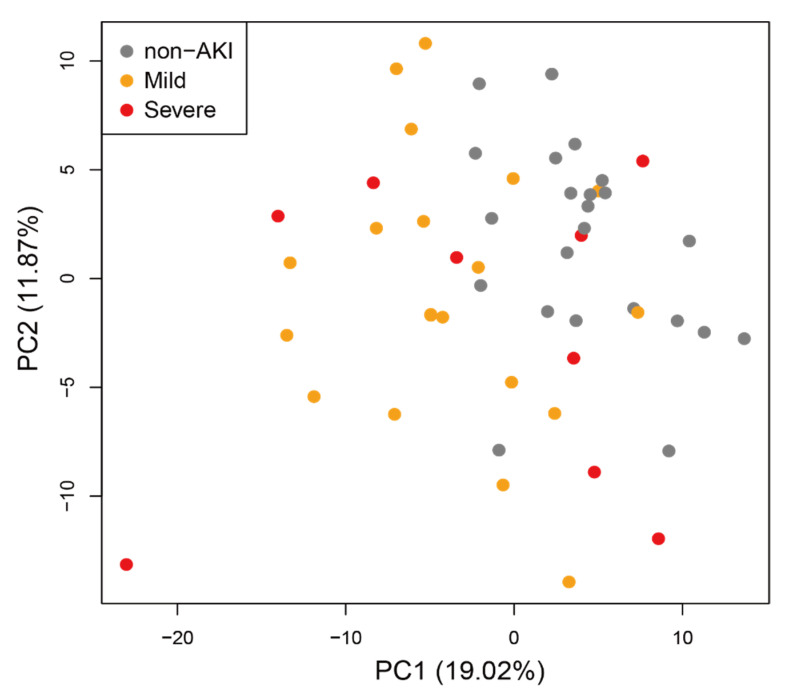
Principal component analysis (PCA) of metabolome profiles in the 1st subject group at 24 h.

**Figure 3 metabolites-11-00671-f003:**
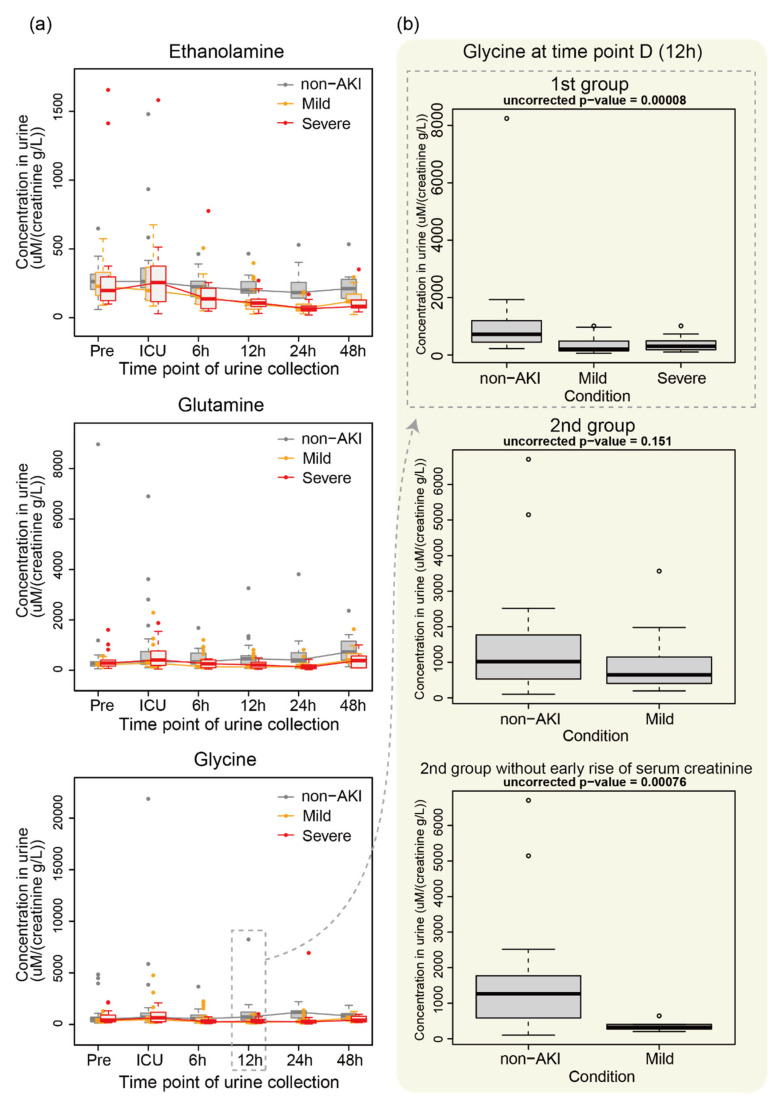
Example of metabolite levels in non-AKI, mild AKI and severe AKI. (**a**) Changes in the levels of ethanolamine, glutamine and glycine over the time points in the 1st group of subjects. (**b**) Glycine level at 12 h in the 1st and 2nd groups (**top** two). The glycine level for subjects in the 2nd group who were not diagnosed with AKI based on serum creatinine within two days of ICU admission is also shown (**bottom**).

**Figure 4 metabolites-11-00671-f004:**
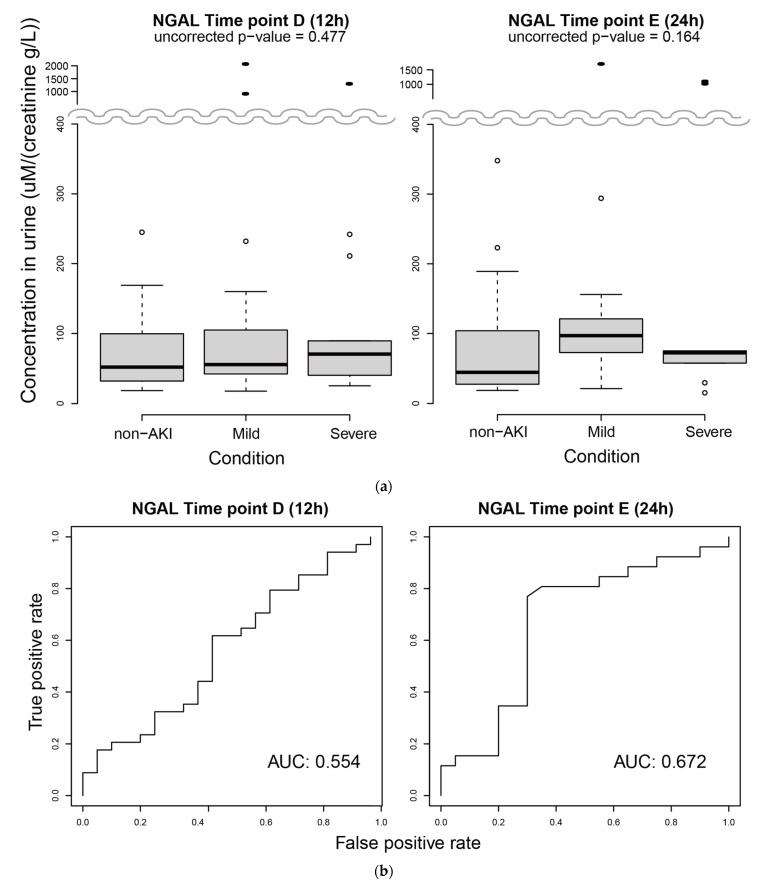
Measurement of urinary NGAL. (**a**) Concentrations of urinary NGAL at 12 h (**left**) and 24 h (**right**). Filled ellipses are extreme outliers. (**b**) Performance assessment of AKI prediction based on the NGAL level at 12 h (**left**) and 24 h (**right**) using ROC curves.

**Figure 5 metabolites-11-00671-f005:**
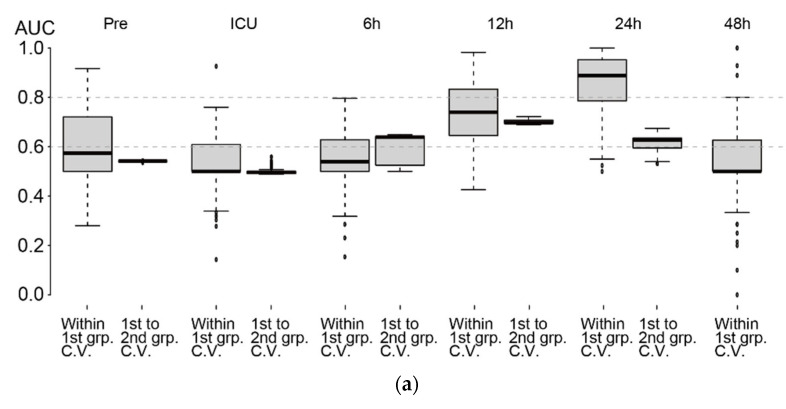
Assessment of AKI prediction using LASSO. (**a**) AKI prediction performances at each time point based on the AUC. AUC of within 1st group cross-validation (**left** side of each boxplot pair) and 1st to 2nd group cross-validation (**right** side of each boxplot pair) are shown. Note that we did not have a sufficient number of samples at 48 h in the 2nd group. (**b**,**c**) (**left**) ROC curves using ¾ of the subjects in the 1st group (randomly selected) as the training set and the rest as the test set. The procedure was repeated 100 times. Overlapped curves are drawn darker. (**right**) Top ten variables based on variable selection rates (designated as V. S. R. on the **top**-**left** of the plot) among 100 iterations of training using ¾ of the 1st group. The distribution of estimated coefficients of the variables is shown. Note that although four clinical variables were included, they were rarely selected by LASSO. (**b**) 12 h, (**c**) 24 h.

**Table 1 metabolites-11-00671-t001:** Top ten metabolites and time points that showed significant differences between non-AKI, mild AKI and severe AKI in the 1st group.

Metabolite Name	Uncorrected *p*-Values	BH Corrected *p*-Values	BF Corrected *p*-Values
E_K: Ethanolamine	5.04 × 10^−6^	0.00769	0.00769
E_K: Gln	6.88 × 10^−5^	0.0347	0.105
D_K: Gly	8.17 × 10^−5^	0.0347	0.125
E_K: 2-Hydroxypentanoate	0.000102	0.0347	0.155
E_K: Gly	0.000114	0.0347	0.173
E_U: AU034 (C9H16N2O4)	0.000178	0.0452	0.271
D_K: Ethanolamine	0.000239	0.0521	0.365
E_U: AU014 (C6H11NO4)	0.00041	0.0603	0.624
E_K: Ser	0.000431	0.0603	0.656
D_K: Succinate	0.000473	0.0603	0.721

The first letter before the metabolite name denotes the time point (D: 12 h, E: 24 h). The second letter denotes whether the metabolite is known (K) or uncharacterized (U). BH: Benjamini–Hochberg, BF: Bonferroni.

**Table 2 metabolites-11-00671-t002:** Top ten significant metabolites from the first mixed effects model with the assumption that the metabolite level is influenced by disease states, time points and subjects.

Metabolite Name	Uncorrected *p*-Values	BH Corrected *p*-Values	BF Corrected *p*-Values
Gly	8.68 × 10^−5^	0.0181	0.0302
NCGC003	0.000118	0.0181	0.0410
NCGA012	0.000189	0.0181	0.0657
Urea	0.000256	0.0181	0.0889
Urate	0.000318	0.0181	0.111
NCGA025	0.000327	0.0181	0.114
Ethanolamine	0.000364	0.0181	0.127
NCGA003	0.000487	0.0212	0.170
Gln	0.000619	0.0222	0.216
N,N-Dimethylglycine	0.000697	0.0222	0.243

Effects of disease states on metabolite levels were tested. BH: Benjamini–Hochberg, BF: Bonferroni.

**Table 3 metabolites-11-00671-t003:** Top 10 significant metabolites from the second model with the assumption that the metabolite level is influenced by disease states, time points and subjects. The magnitude of metabolite changes over the time points is also assumed to be influenced by the disease state.

Metabolite Name	Uncorrected *p*-Values	BH Corrected *p*-Values	BF Corrected *p*-Values
Piperidine	2.54 × 10^−6^	0.000883	0.000883
AU035 (C7H8O6S)	1.04 × 10^−5^	0.00181	0.00363
CU021 (C6H10N2O4)	6.85 × 10^−5^	0.00794	0.0238
CU001 (C4H9N)	0.000110	0.00953	0.0381
CU043 (C14H22N2O)	0.000191	0.0133	0.0663
NCGC008	0.000412	0.0239	0.143
Taurine	0.000635	0.0300	0.221
Methanesulfonate	0.000689	0.0300	0.240
AU021 (C7H8O4S)	0.00131	0.0507	0.456
3-Hydroxykynurenine	0.00184	0.0583	0.639

Effects of disease states on the changes in metabolite levels over the time points were tested. BH: Benjamini–Hochberg, BF: Bonferroni.

**Table 4 metabolites-11-00671-t004:** Clinical characteristics of the 1st group of subjects.

Characteristic	Non-AKI (*n* = 23)	Mild AKI (*n* = 24)	Severe AKI (*n* = 14)
Gender [Male (M), Female (F)]	M: 16 (70%) F: 7 (30%)	M: 19 (79%) F: 5 (21%)	M: 9 (64%) F: 5 (36%)
Age range (median)	52–83 (68.0)	47–84 (71.0)	53–76 (67.0)
Department			
Cardiology	12 (52%)	10 (41%)	4 (28%)
Gastroenterology	7 (30%)	9 (37%)	6 (42%)
Breast	1 (4%)	0 (0%)	1 (7%)
Orthopedics	1 (4%)	1 (4%)	0 (0%)
Respiratory Medicine	2 (8%)	1 (4%)	0 (0%)
Transplantation	0 (0%)	0 (0%)	1 (7%)
Vascular	0 (0%)	3 (12%)	1 (7%)
Urology	0 (0%)	0 (0%)	1 (7%)

**Table 5 metabolites-11-00671-t005:** Clinical characteristics of the 2nd group of subjects.

Characteristic	Non-AKI (*n* = 40)	Mild AKI (*n* = 20)
Gender [Male (M), Female (F)]	M: 34 (85%) F: 6 (15%)	M: 19 (95%) F: 1 (5%)
Age range (median)	40–83 (68.0)	31–81 (72.5)
Department		
Cardiology	14 (35%)	3 (15%)
Gastroenterology	7 (17%)	11 (55%)
Neurology	6 (15%)	1 (5%)
Orthopedics	1 (2%)	0 (0%)
Respiratory Medicine	2 (5%)	2 (10%)
Transplantation	1 (2%)	1 (5%)
Vascular	4 (10%)	2 (10%)
Oral	2 (5%)	0 (0%)
Otorhinolaryngology	3 (7%)	0 (0%)

## Data Availability

Due to protect patient privacy, patient data in this study may be available after reviewing the purpose and plan by the study committee. Requests must be sent to the corresponding author.

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
