# Peer review of "Urinary Metabolome Analyses of Patients with Acute Kidney Injury Using Capillary Electrophoresis-Mass Spectrometry"

_metabolites, 2021, doi:10.3390/metabo11100671_

Round 1
Reviewer 1 Report
Please add a short paragraph in the introduction about the use of CE/MS for metabolomic studies, you should cite your own work and the article ta et al Analytica chimica acta 2021 DOI: 10.1016/j.aca.2021.338233
line 73 change "After admission, 24 and 14 developed..."
The main question is what CE/MS-metabolomic will present as results for Acute kidney injury.
The method (CE/MS) is relatively new and was never used for AKI. The results show some biomarkers characteristic of the patology.
On AKI it was the firts time that CE/MS metabolomic was used. Mainly because it is the most performant lab in the world using this technic.
The paper is well written. The text is clear and easy to read. The conclusions are consistent with the evidence and arguments presented. They address the main question posed.
The manuscript lakcs in presenting the use of CE/MS for metabolomic studies.
Author Response
Comment 1-1: Please add a short paragraph in the introduction about the use of CE/MS for metabolomic studies. / The manuscript lacks in presenting the use of CE/MS for metabolomic studies.
Response 1-1: We added some more introduction about the use of CE-MS for metabolomics studies (page 2 lines 58-65).
Comment 1-2: The main question is what CE/MS-metabolomic will present as results for Acute kidney injury. The method (CE/MS) is relatively new and was never used for AKI. The results show some biomarkers characteristic of the pathology. On AKI it was the first time that CE/MS metabolomic was used. Mainly because it is the most performant lab in the world using this technic.
Response 1-2: We added some advantages of using CE-MS to analyze metabolomic profiles of patients with AKI, compared to previous studies using LC-MS (page 12 the last paragraph).
Comment 1-3: You should cite your own work and the article ta et al Analytica chimica acta 2021 DOI: 10.1016/j.aca.2021.338233
Response 1-3: The following citation suggested by the reviewer was added to the manuscript (page 2 line 61):
Ta HY, Collin F, Perquis L, Poinsot V, Ong-Meang V, Couderc F (2021) Twenty years of amino acid determination using capillary electrophoresis: A review. Anal Chim Acta 1174:338233 (PMID: 34247732)
Comment 1-4: line 73 change "After admission, 24 and 14 developed..."
Response 1-4: The authors changed the statement from
“After admission, twenty-four and 14 developed mild and severe AKI, respectively (designated as mild AKI and severe AKI).”
to
“After admission, 24 and 14 developed mild and severe AKI, respectively (designated as mild AKI and severe AKI).”
according to reviewer’s comment.
Reviewer 2 Report
Early detection of acute kidney incury (AKI) with adequate biomarkers is an essential issue for clinicians and still remains difficult. In their study, Saito et al. focused on patients admitted to an intensive care unit (ICU) after surgery. They conducted a comprehensive metabolic profiling of urine samples from patients with AKI and used the capillary electrophoresis time-of-flight mass spectrometry (CE-TOFMS).
This method is a really an innovative laboratory method and is more sensitive compared to LC-MS. Therefore, this work has some merit. However, some minor points should be considered before the manuscript can be published:
- Figure 1: (a) and (b) should be clearly marked in this figure.
- Tables 1-3: p-values should be given with maximal 3 decimal numbers
- Alle p-values during the whole manuscript should be given with maximal 3 decimal numbers, only.
- Conclusio: The authors designed a very laborious study. In their conclusion they should clearly state, which additional value they derive from their big data analyses!
Author Response
Comment 2-1: Figure 1: (a) and (b) should be clearly marked in this figure.
Response 2-1: Yes, I noticed that mark (a) has been dropped at some point. I have added mark (a) to Figure 1.
Comment 2-2: Tables 1-3: p-values should be given with maximal 3 decimal numbers. All p-values during the whole manuscript should be given with maximal 3 decimal numbers, only.
Response 2-2: Revised accordingly. P-value representations in Tables 1-3, Figures 3b and 4a have been changed.
Comment 2-3: Conclusion: The authors designed a very laborious study. In their conclusion they should clearly state, which additional value they derive from their big data analyses!
Response 2-3: In the conclusion, we stressed that we dealt with big size of the metabolome data and, from the data, efficiently identified metabolites and time points that show patterns specific to patients who develop AKI, using our statistical models and machine learning technique.